# Acute Kidney Injury in Patients Undergoing Cardiac Transplantation: A Meta-Analysis

**DOI:** 10.3390/medicines6040108

**Published:** 2019-11-01

**Authors:** Charat Thongprayoon, Ploypin Lertjitbanjong, Panupong Hansrivijit, Anthony Crisafio, Michael A. Mao, Kanramon Watthanasuntorn, Narothama Reddy Aeddula, Tarun Bathini, Wisit Kaewput, Wisit Cheungpasitporn

**Affiliations:** 1Division of Nephrology and Hypertension, Mayo Clinic, Rochester, MM 55905, USA; charat.thongprayoon@gmail.com; 2Department of Internal Medicine, Bassett Medical Center, Cooperstown, NY 13326, USA; ploypinlert@gmail.com (P.L.); kanramon@gmail.com (K.W.); 3Department of Internal Medicine, University of Pittsburgh Medical Center Pinnacle, Harrisburg, PA 17105, USA; p.hansrivijit@gmail.com; 4St George’s University, School of Medicine University Centre Grenada, West Indies, St George, Grenada; anthony.crisafio@gmail.com; 5Division of Nephrology and Hypertension, Mayo Clinic, Jacksonville, FL 32224, USA; mao.michael@mayo.edu; 6Division of Nephrology, Deaconess Health System, Evansville, IN 47747, USA; dr.anreddy@gmail.com; 7Department of Internal Medicine, University of Arizona, Tucson, AZ 85721, USA; tarunjacobb@gmail.com; 8Department of Military and Community Medicine, Phramongkutklao College of Medicine, Bangkok 10400, Thailand; wisitnephro@gmail.com; 9Division of Nephrology, University of Mississippi Medical Center, Jackson, MS 39216, USA

**Keywords:** AKI, acute kidney injury, epidemiology, heart transplantation, cardiac transplantation

## Abstract

**Background:** Acute kidney injury (AKI) is a common complication following solid-organ transplantation. However, the epidemiology of AKI and mortality risk of AKI among patients undergoing cardiac transplantation is not uniformly described. We conducted this study to assess the incidence of AKI and mortality risk of AKI in adult patients after cardiac transplantation. **Methods:** A systematic review of EMBASE, MEDLINE, and Cochrane Databases was performed until June 2019 to identify studies evaluating the incidence of AKI (by standard AKI definitions), AKI requiring renal replacement therapy (RRT), and mortality risk of AKI in patients undergoing cardiac transplantation. Pooled AKI incidence and mortality risk from the included studies were consolidated by random-effects model. The protocol for this study is registered with PROSPERO (no. CRD42019134577). **Results:** 27 cohort studies with 137,201 patients undergoing cardiac transplantation were identified. Pooled estimated incidence of AKI and AKI requiring RRT was 47.1% (95% CI: 37.6–56.7%) and 11.8% (95% CI: 7.2–18.8%), respectively. The pooled ORs of hospital mortality and/or 90-day mortality among patients undergoing cardiac transplantation with AKI and AKI requiring RRT were 3.46 (95% CI, 2.40–4.97) and 13.05 (95% CI, 6.89–24.70), respectively. The pooled ORs of 1-year mortality among patients with AKI and AKI requiring RRT were 2.26 (95% CI, 1.56–3.26) and 3.89 (95% CI, 2.49–6.08), respectively. **Conclusion:** Among patients undergoing cardiac transplantation, the incidence of AKI and severe AKI requiring RRT are 47.1% and 11.8%, respectively. AKI post cardiac transplantation is associated with reduced short term and 1-year patient survival.

## 1. Introduction 

Acute kidney injury (AKI), a heterogeneous clinical syndrome, is a significant health problem worldwide with a steady increase in the incidence [1]. The burden of AKI is 13.3 million cases a year globally and is associated with high mortality at 1.7 million deaths annually [2,3,4]. Survivors of AKI are subsequently at higher risk for cardiovascular events, chronic kidney disease (CKD), and progression to end-stage kidney disease (ESKD) [5]. The number of hospitalizations from AKI is steeply rising as evidenced by the national inpatient sample data in the United States between 2000 onwards with close to 3.9 million admissions in 2014 [6].

There is a steady and gradual annual global increase in the number of cardiac transplantations ever since the first successful transplantation in 1967 [7]. The cardiac transplantation has evolved over the years and ever since the introduction of the newer immunosuppressive calcineurin inhibitors, it has evolved into a life-sustaining treatment of choice for the many end-stage heart disease patients [8,9,10,11,12,13,14]. However, AKI after cardiac transplantation is a common complication, which can lead to subsequent progressive CKD and ESKD requiring dialysis post cardiac transplantation [15,16,17]. Despite tremendous advances in the field of cardiac transplant medicine and surgical techniques, there is a paucity of data on the incidence, associated risk factors, and mortality risk of AKI in patients undergoing cardiac transplantation. In addition, the incidence of AKI among patients undergoing cardiac transplantation is not uniformly described among studies, ranging from 40% to 70% [18,19,20,21,22,23].

Thus, this meta-analysis aims to assess the incidence of AKI and mortality risk of AKI in adult patients after cardiac transplantation.

## 2. Methods

### 2.1. Search Strategy

The protocol for this study is registered with PROSPERO (International Prospective Register of Systematic Reviews no. CRD42019134577). We conducted a systematic literature review of Ovid MEDLINE, EMBASE, and the Cochrane Database of Systematic Reviews until June 2019 to evaluate the incidence of AKI and mortality risk of AKI in adult patients undergoing cardiac transplantation. Independent reviewers (C.T. and P.L.) conducted a systematic literature search using a search strategy that incorporated the search terms “heart” OR “cardiac” AND “transplant” OR “transplantation” AND “acute renal failure.” OR “acute kidney injury”. Additional details on the search approach employed for each database is provided in Appendix A. A manual search for conceivably related articles utilizing references of the included studies was additionally performed. No language restriction was implemented. This systematic review was conducted following the PRISMA (Preferred Reporting Items for Systematic Reviews and Meta-Analysis) [24].

### 2.2. Study Selection

Studies were eligible for this meta-analysis if the studies were clinical trials or observational studies that had data on the incidence of AKI (using standard AKI definitions including Kidney Disease: Improving Global Outcomes (KDIGO) classifications [25], Acute Kidney Injury Network (AKIN) [26], Risk, Injury, Failure, Loss of kidney function, and End-stage kidney disease (RIFLE) [27]), and AKI requiring renal replacement therapy (RRT), and mortality risk of AKI in patients (aged 18 years and older) undergoing cardiac transplantation. Included studies had to have the data to assess the incidence or mortality risk of AKI. Retrieved studies were individually evaluated for eligibility by the two investigators (C.T. and P.L.). Conflicts were discussed and solved by consensus or by a third reviewer (W.C.).

A structured information collecting form was utilized to gather the following information from each included study including title of the article, name of investigators, year of the study, country where the study was conducted, publication year, incidence of AKI, definition of AKI, risk factor for AKI, and mortality risk of AKI in patients undergoing cardiac transplantation.

### 2.3. Statistical Analysis

We used comprehensive meta-analysis software version 3.3.070 (Biostat Inc, United States) for all analyses. Pooled AKI incidence and mortality risk of included studies were incorporated by the generic inverse variance method of DerSimonian-Laird, which indicated the weight of each study depending on its variance [28]. Due to the likelihood of inter-observation variance, we utilized a random-effects model for meta-analyses of the incidence and mortality risk of AKI among patients undergoing cardiac transplantation. Statistical heterogeneity of studies was evaluated by the Cochran’s Q test (statistically significant as *p* < 0.05) and the *I*^2^ statistic (≤ 25% represents insignificant heterogeneity, 26% to 50% represents low heterogeneity, 51% to 75% represents moderate heterogeneity, and ≥ 75 % represents high heterogeneity) [29]. Publication bias was assessed by both the Egger test and the funnel plot [30].

## 3. Results

A total of 4252 conceivably suitable articles were initially identified with our search strategy. Subsequently, 1678 articles were excluded due to being duplicates, and we also excluded 2367 articles that were either in-vitro studies, animal studies, pediatric patient population, correspondences, review articles, or case reports. Thus, 207 articles were entered for full-length article review. Subsequently, 94 studies were excluded as these studies did not have data on the incidence or mortality of AKI, 77 articles were additionally excluded because they were not observational studies or clinical trials, and 9 studies were excluded because they did not utilize a standard AKI definition or did not describe the incidence of AKI requiring RRT. Finally, 27 cohort studies [8,18,19,20,31,32,33,34,35] with 137,201 patients undergoing cardiac transplantation were identified. The flowchart of this study is shown in Figure 1. The characteristics of the included studies are shown in Table 1.

### 3.1. Incidence of AKI Among Patients Undergoing Cardiac Transplantation

Pooled estimated incidence of AKI was 47.1% (95% confidence intervals (CI): 37.6–56.7%, *I*^2^ = 97%, Figure 2) and severe AKI requiring RRT among patients undergoing cardiac transplantation was 11.8% (95% CI: 7.2%–18.8%, *I*^2^ = 98%, Figure 3). Subgroup analyses were conducted according to AKI definitions. The pooled incidence of AKI by RIFLE, AKIN, and KDIGO criteria were 35.3% (95% CI: 20.5–53.5%, *I*^2^ = 95%), by AKIN criteria: 29.9% (95% CI: 16.3–48.3%, *I*^2^ = 92%), and KGDIGO criteria: 62.8% (95% CI: 49.2–74.7%, *I*^2^ = 96%). 

Subgroup analyses based on year of study (before and after the year 2015) were performed to assess if there was any difference in the incidence of AKI among studies from the recent years vs the former years. The pooled estimated incidence rates of AKI and AKI requiring RRT before the year of 2015 were 33.8% (95% CI: 18.8–52.9%, *I*^2^ = 96%) and 9.4% (95% CI: 5.6–15.3%, *I*^2^ = 61%), respectively. The pooled estimated incidence rates of AKI and AKI requiring RRT after the year of 2015 were 49.4% (95% CI: 35.7–63.2%, *I*^2^ = 97%) and 12.3% (95% CI: 7.2–20.1%, *I*^2^ = 98%), respectively.

### 3.2. Impact of AKI on Mortality among Patients Undergoing Cardiac Transplantation

Mortality risk associated with AKI among patients undergoing cardiac transplantation is summarized in Table 2. Pooled ORs of hospital mortality and/or 90-day mortality among patients undergoing cardiac transplantation with AKI and severe AKI requiring RRT were 3.46 (95% CI, 2.40–4.97, *I*^2^ = 0%, Figure 4A) and 13.05 (95% CI, 6.89–24.70, *I*^2^ = 75%, Figure 4B), respectively. When the analysis was restricted to only studies with confounder-adjusted analysis, the higher hospital mortality was still significant in patients undergoing cardiac transplantation with AKI pooled OR of 4.10 (95% CI, 2.57–6.54, *I*^2^ = 0%, Appendix A) and severe AKI requiring RRT with pooled OR of 8.93 (95% CI, 3.48–22.92, *I*^2^ = 57%, Appendix A). 

The pooled ORs of 1-year mortality among patients undergoing cardiac transplantation with AKI were 2.26 (95% CI, 1.56–3.26, *I*^2^ = 0%, Appendix A) and AKI requiring RRT were 3.89 (95% CI, 2.49–6.08, *I*^2^ = 17%, Appendix A). When the meta-analysis was restricted to studies with confounder-adjusted analysis, the higher 1-year mortality was still significant in patients undergoing cardiac transplantation with AKI (pooled ORs of 3.11 (95% CI, 1.66–5.82, *I*^2^ = 0%, Appendix A)) and AKI requiring RRT with pooled ORs of 4.06 (95% CI, 1.69–9.75, *I*^2^ = 36%, Appendix A). Meta-regression demonstrated that year of the study was not correlated with the risks of hospital mortality (and/or 90-day mortality) (*p* = 0.93) or 1-year mortality (*p* = 0.44) among patients undergoing cardiac transplantation with AKI. 

### 3.3. Evaluation for Publication Bias

Funnel plots (Appendix A) and Egger’s regression asymmetry tests were used to evaluate publication bias in meta-analyses assessing hospital mortality (and/or 90-day mortality) and 1-year mortality of AKI among patients undergoing cardiac transplantation. We found no publication bias for analyses assessing the hospital mortality (and/or 90-day mortality; *p* = 0.46) and 1-year mortality of AKI among patients undergoing cardiac transplantation (*p* = 0.24). 

## 4. Discussion

In this study, we revealed that AKI and AKI requiring RRT among patients undergoing cardiac transplantation are fairly common (47% and 12%, respectively). In addition, the incidence of AKI and severe AKI requiring RRT have been increasing in recent years. AKI post cardiac transplantation is associated with increased short term (3.5-fold) and 1-year (2.3-fold) mortality. Despite advances in transplant medicine, meta-regression showed that the risks of hospital mortality (and/or 90-day mortality) or 1-year mortality among patients undergoing cardiac transplantation with AKI has not improved over time.

As there are currently no effective targeted pharmacotherapies available for AKI, prevention of AKI and early identification of patients at risk for AKI among patients undergoing heart transplantation may potentially play an important role in improving patient outcomes, given high short- and long-term mortality risks associated with AKI after cardiac transplantation. Reported risk factors for AKI in patients undergoing cardiac transplantation are shown in Table 3. Preoperative risk factors including chronic kidney disease (CKD) [8,19,36,43,51], diabetes mellitus (DM) [36,43], and older age [19,41,52]. were consistently demonstrated as associated risk factors for AKI in patients after cardiac transplantation. Perioperative and post risk factors for AKI among patients undergoing cardiac transplantation included increased cardiopulmonary bypass and surgery time [20,36,50], supratherapeutic calcineurin inhibitor concentration [38,41,50], post-operative bleeding, anemia, and blood transfusion [18,20,39,52], postoperative RV failure and the need for a right ventricular assist device (RVAD) after heart transplant [8,23], duration of mechanical ventilation and the need for venoarterial extra-corporal membrane oxygenation (VA ECMO) after heart transplant [23].

At the patient level, the primary indications for cardiac transplantation are nonischemic cardiomyopathy (53%) and ischemic cardiomyopathy (38%) [54]. These patients generally carry significant co-morbidities, such as diabetes, hypertension, and congestive heart failure (right heart failure and reduced pulmonary artery pulsatility index prior to heart transplantation) that predispose them to develop AKI [18]. At the peri-operative level, AKI among patients undergoing cardiac transplantation has been reportedly associated with duration of the cardiopulmonary bypass (CPB), prolonged mechanical ventilation, ischemic reperfusion injury, and anemia [8,20,38,41,53,55,56]. The use of CPB has been associated with AKI risk, which is supported by a meta-analysis from Pickering et al. [57]. Moreover, CPB duration, the use of mannitol and furosemide during CPB, and urine output during CPB can influence the incidence of AKI [58].

There are some limitations that are worth noting here. Most included studies used serum creatinine as a criterion for AKI, which might underestimate the incidence of AKI. Moreover, our meta-analysis primarily depends on observational studies given the lack of clinical trials on this particular topic. Thus, it is difficult to conclude a causal relationship based on data from observation studies. Nonetheless, more in-depth data from population-based studies or clinical trials on AKI prevention among patients receiving cardiac transplantation is practical and encouraged. For instance, a recent single-center prospective cohort study has demonstrated some benefit of utilizing novel biomarkers in predicting post-operative AKI after cardiac surgery (n = 23) [59]. A larger cohort is currently under investigation. Lastly, AKI is a known independent risk factor for the development of CKD [60,61,62] and one of the important adverse effects of calcineurin inhibitors among cardiac transplant patients is the development of CKD leading to ESKD. Future studies are required if optimizing/modifying the immunosuppressive regimen among cardiac transplant patients with AKI can help prevent CKD and delay CKD progression.

In summary, we revealed that almost half of patients undergoing cardiac transplantation developed AKI. Severe AKI requiring RRT is as high as 12%. AKI and AKI requiring RRT after cardiac transplantation are associated with increased risk of 90-day mortality and 1-year mortality. We also emphasized that AKI requiring RRT is a poor prognostic predictor as these patients have a 13-fold increased risk of dying within 90 days post-transplant.

## Figures and Tables

**Figure 1 medicines-06-00108-f001:**
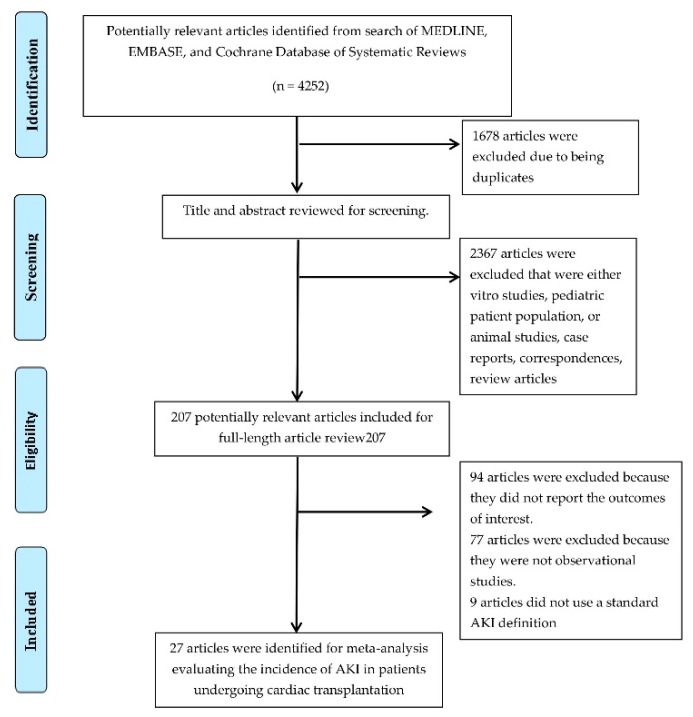
The flowchart for the systematic review.

**Figure 2 medicines-06-00108-f002:**
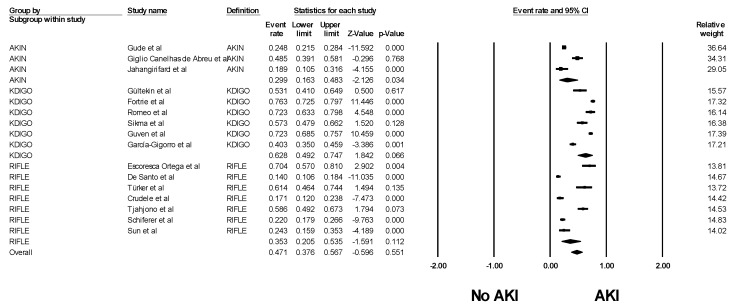
Forest plots of the included studies evaluating incidence of AKI among patients undergoing cardiac transplantation. A diamond data marker represents the overall rate from individual study (square data marker) and 95% CI.

**Figure 3 medicines-06-00108-f003:**
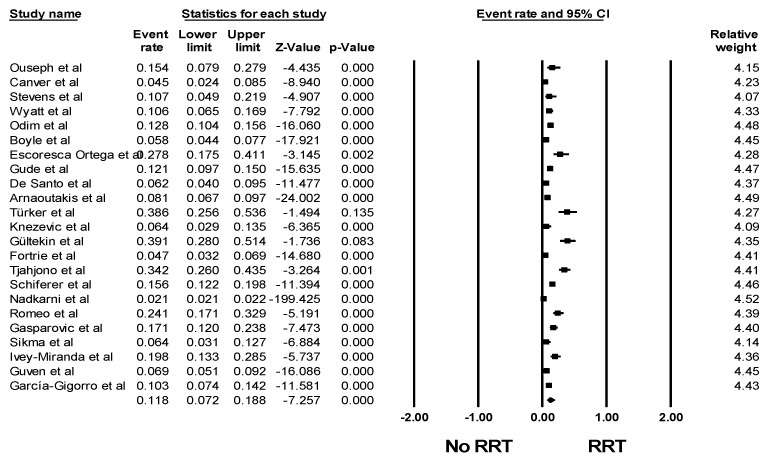
Forest plots of the included studies assessing incidence rates of AKI requiring RRT among patients undergoing cardiac transplantation. A diamond data marker depicts the overall rate from each included study (square data marker) and 95% CI.

**Figure 4 medicines-06-00108-f004:**
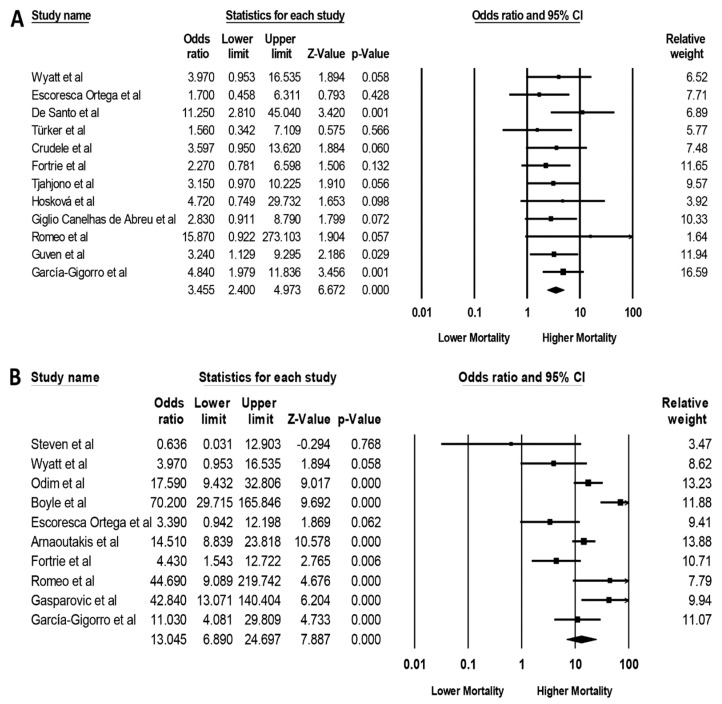
Forest plots of the included studies assessing (**A**) hospital mortality and/or 90-day mortality among patients undergoing cardiac transplantation with AKI and (**B**) hospital mortality and/or 90-day mortality among patients undergoing cardiac transplantation with AKI on RRT. A diamond data label serves as the overall rate from each included study (square data marker) and 95% CI.

**Table 1 medicines-06-00108-t001:** Included studies in this systematic review of AKI incidence and mortality in patients undergoing cardiac transplantation. [8,18,19,20,31,32,33,34,35].

Study	Year	Country	Patient Population	Number	Definition of AKI	Incidence of AKI	Mortality
Ouseph et al. [31]	1998	USA	Orthotopic heart transplant	52	RRT	RRT, 8/52 (15.4%)	1-year mortality RRT, 5/8 (62.5%)
Canver et al. [32]	2000	USA	Orthotopic heart transplant	199	RRT	RRT, 9/199 (4.5%)	RRT, 4/9 (44.4%)
Stevens et al. [33]	2004	Canada	Heart transplant	56	CRRT	CRRT, 6/56 (10.71%)	RRT, 0/6 (0%)
Wyatt et al. [34]	2004	USA	Heart transplant in New York State	141	RRT	RRT, 15/141 (10.6%)	N/A
Odim et al. [35]	2005	USA	Orthotopic heart transplant	627	RRT	RRT, 80/627 (12.8%)	Hospital mortality RRT, 33/80 (41.3%)
Boyle et al. [36]	2006	USA	Orthotopic heart transplant	756	RRT	RRT, 44/756 (5.8%)	Hospital mortality RRT, 22/44 (50%) 30-day mortality RRT, 17/44 (38.6%)
Escoresca Ortega et al. [37]	2010	Spain	Heart transplant	54	RIFLE criteria	AKI, 38/54 (70.4%) RRT, 15/54 (27.8%)	AKI, 30/38 (78.94%) RRT, 7/15 (46.7%)
Gude et al. [38]	2010	Norway	Orthotopic heart transplant	585	AKIN criteria	AKI, 145/585 (24.8%) RRT, 71/585 (12.1%)	30-day mortality RRT, 11/71 (15.5%) 90-day mortality RRT, 15/71 (21.1%)
De Santo et al. [39]	2011	Italy	Orthotopic heart transplant	307	RIFLE criteria	AKI, 43/307 (14.0%) CVVH, 19/307 (6.2%)	Hospital mortality AKI, 12/43 (27.9%) 1-year mortality AKI, 12/43 (27.9%)
Arnaoutakis et al. [40]	2012	USA	LVAD-bridged heart transplant from UNOS data	1312	RRT	RRT, 106/1312 (8.1%)	90-day mortality RRT, 41/93 (44.1%)
Türker et al. [41]	2013	Turkey	Heart transplant	44	RIFLE criteria	AKI, 27/44 (61.4%) RRT, 17/44 (38.6%)	AKI, 7/26 (26.9%)
Crudele et al. [42]	2013	Italy	Heart transplant	158	RIFLE criteria	AKI, 27/158 (17.1%)	AKI, 9/27 (33.3%)
Knezevic et al. [43]	2014	Slovenia	Heart transplant	94	RRT	RRT, 6/94 (6.4%)	N/A
Gültekin et al. [44]	2015	Turkey	Orthotopic heart transplant	64	KDIGO criteria	AKI, 34/64 (53.1%) RRT, 25/64 (39.1%)	N/A
Fortrie et al. [8]	2016	The Netherlands	Heart transplant	531	KDIGO criteria	AKI, 405/531 (76.3%) RRT, 25/531 (4.7%)	Hospital mortality AKI, 28/405 (6.9%) RRT, 5/25 (20.0%) 1-year mortality AKI, 41/405 (10.12%) RRT, 7/25 (28.0%)
Tjahjono et al. [20]	2016	Australia	Orthotopic heart transplant	111	RIFLE criteria	AKI, 65/111 (58.6%) RRT, 38/111 (34.2%)	AKI, 15/65 (23.1%)
Schiferer et al. [45]	2016	Austria	Heart transplant	346	RIFLE criteria	AKI RIFLE, 76/346 (22.0%) RRT, 54/346 (15.6%)	1-year mortality RIFLE AKI, 19/76 (25.00%) RRT, 20/54 (37.0%)
Nadkarni et al. [46]	2017	USA	Orthotopic heart transplant from the Nationwide Inpatient Sample database during 2002–2013	130,143	RRT	RRT, 2776/130,143 (2.1%)	N/A
Giglio Canelhas de Abreu et al. [47]	2017	Brazil	Heart transplant	103	AKIN criteria	AKI, 50/103 (48.5%)	AKI, 16/50 (32%)
Sun et al. [48]	2018	USA	Patients with preexisting LVADs who underwent orthotopic heart transplantation	74	RIFLE criteria	AKI, 18/74 (24.3%)	N/A
Romeo et al. [19]	2018	Argentina	Heart transplant	112	KDIGO criteria	AKI, 81/112 (72.3%) RRT, 27/112 (24.1%)	Hospital mortality AKI, 16/81 (19.8%) RRT, 14/27 (51.9%) 1-year mortality AKI, 19/81 (23.5%) RRT, 16/27 (59.3%)
Gašparović et al. [49]	2018	Croatia	Heart transplant	158	RRT	RRT, 27/158 (17.1%)	3-month mortality RRT 17/27 (63.0%)
Sikma et al. [50]	2018	The Netherlands	Heart transplant	110	KDIGO criteria	AKI, 63/110 (57.3%) RRT, 7/110 (6.4%)	N/A
Ivey-Miranda et al. [51]	2018	Mexico	Heart transplant	106	RRT	RRT, 21/106 (19.81%)	N/A
Jahangirifard et al. [52]	2018	Iran	Heart transplant	53	AKIN criteria	AKI, 10/53 (18.87%)	N/A
Guven et al. [53]	2018	The Netherlands	Heart transplant	595	KDIGO criteria	AKI, 430/595 (72.3%) RRT, 41/595 (6.9%)	Hospital mortality AKI, 32/430 (7.4%) 1-year mortality AKI, 43/430 (10.0%) RRT, 9/41 (22.0%)
García-Gigorro et al. [18]	2018	Spain	Heart transplant	310	KDIGO criteria	AKI, 125/310 (40.3%) RRT, 32/310 (10.3%)	Hospital mortality AKI, 20/125 (16%) RRT, 15/32 (46.9%)

Abbreviations: AKIN, Acute Kidney Injury Network; CRRT, continuous renal replacement therapy; ICU, intensive care unit; KDIGO, Kidney Disease Improving Global Outcomes; N/A, not available; LVAD, left ventricular assist device; RIFLE, Risk, Injury, Failure, Loss of kidney function, and End-stage kidney disease; RRT, Renal replacement therapy; UNOS, United Network for Organ Sharing; USA, United States of America.

**Table 2 medicines-06-00108-t002:** Included studies in this analysis of AKI associated mortality risk in patients undergoing cardiac transplantation.

Study	Year	OR for Mortality	Confounder Adjustment
Ouseph et al. [31]	1998	1-year mortality RRT: 16.67 (2.86–97.09)	None
Steven et al. [33]	2004	0/6 in RRT vs 5/50 (10%) in non-RRT	None
Wyatt et al. [34]	2004	Hospital mortality AKI, 3.97 (0.95–16.48) RRT, 8.96 (1.75–45.80)	Age, sex, race, DM, transplant center
Odim et al. [35]	2006	Hospital mortality RRT, 17.59 (9.43–32.80)	None
Boyle et al. [36]	2006	Hospital mortality RRT, 70.20 (29.71–165.82)	None
Escoresca Ortega et al. [37]	2010	AKI, 1.70 (0.46–6.34) RRT, 3.39 (0.94–12.17)	None
De Santo et al. [39]	2011	Hospital mortality AKI, 11.25 (2.81–45.04) 1-year mortality AKI, 4.25 (1.41–12.79)	Preoperative GFR, infection, graft failure
Arnaoutakis et al. [40]	2012	90-day mortality RRT, 14.51 (8.84–23.82)	None
Türker et al. [41]	2013	AKI, 1.56 (0.34–7.06)	None
Crudele et al. [42]	2013	Mortality AKI, 3.597 (0.95–13.62)	Recipient age, troponin, reoperation, ischemia duration, HLA mismatch
Fortrie et al. [8]	2016	Hospital mortality AKI, 2.27 (0.78–6.59) RRT, 4.43 (1.54–12.70) 1-year mortality AKI, 2.25 (0.93–5.44) RRT, 2.75 (1.13–6.63)	Age, urgency status on waiting list, RV failure, reoperation, primary graft failure, other transplant complications
Tjahjono et al. [20]	2016	AKI, 3.15 (0.97–10.22)	None
Schiferer et al. [45]	2016	1-year mortality RIFLE AKI, 1.86 (1.01–3.45) RRT, 3.71 (1.94–7.07)	None
Hošková et al. [23]	2016	30-day mortality AKI, 4.72 (0.75–29.76) 1-year mortality AKI, 1.52 (0.42–5.46)	None
Giglio Canelhas de Abreu et al. [47]	2017	AKI, 2.83 (0.91–8.78)	Hemoglobin, creatinine, intubation duration, INTERMACS score
Romeo et al. [19]	2018	Hospital mortality 16/81 in AKI vs 0/31 in no AKI RRT, 44.69 (9.09–219.77) 1-year mortality AKI, 4.44 (0.97–20.36) RRT, 6.79 (2.25–20.52)	Age, diabetes, bypass time
Gašparović et al. [49]	2018	3-month mortality RRT, 42.84 (13.07–140.39)	None
Guven et al. [53]	2018	Hospital mortality AKI, 3.24 (1.13–9.30) 1-year mortality AKI, 2.1806 (1.00–4.74) RRT, 2.82 (1.28–6.24)	None
García-Gigorro et al. [18]	2018	Hospital mortality AKI, 4.84 (1.98–11.84) RRT, 11.03 (4.08–29.8)	Acute right ventricular failure, primary graft failure

Abbreviations: AKI, acute kidney injury; GFR, glomerular filtration rate; HLA, human leukocyte antigen; RV, right ventricular.

**Table 3 medicines-06-00108-t003:** Reported Risk Factors for AKI among Patients Undergoing Cardiac Transplantation.

Risk Factors for AKI
**Preoperative risk factors** Higher preoperative serum creatinine or CKD [8,19,36,43,51] Lower serum albumin level [36] Diabetes mellitus [36,43] Increased donor age [38] Older age [19,41,52] Higher BMI [8] Hypertension [46] Higher Logistic EuroSCORE [18] Sepsis [46] Liver disease [46] Elevated troponin I [39] Previous cardiac operation [39] **Perioperative/Postoperative risk factors** Increased cardiopulmonary bypass time [20,36] Increased surgery time [50] Administration of intravenous cyclosporin immediately post-operation, use of cyclosporine [38,41]; supratherapeutic tacrolimus trough concentration [50] Increased graft ischemic time [39] Lower hemoglobin/hematocrit/platelet count [52]; higher blood and cryoprecipitate transfusions [20,39] Post-operative bleeding with subsequent surgical re-exploration [18,20] Postoperative RV failure [8,18] RVAD/VA ECMO after heart transplant [23] Mechanical ventilation [46] Increased peripheral vascular resistance [19] Nadir oxygen delivery < 300 mL/min/m^2^ [51] Higher right atrial pressure, lower pulmonary artery pulsatility index [53] Cardiac tamponade [18]

Abbreviations: CKD, chronic kidney disease; BMI, body mass index; RVAD, right ventricular assist device; VA ECMO, venoarterial extra-corporal membrane oxygenation.

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
