# Peer review of "Acute Kidney Injury in Patients Undergoing Cardiac Transplantation: A Meta-Analysis"

_medicines, 2019, doi:10.3390/medicines6040108_

Round 1

Reviewer 1 Report

As acute kidney injury after heart transplantation is a severe problem with impact on mortality and outcome, the study addresses an important topic. The integration of a number of singular clinical trials into a meta-analysis can improve the understanding of epidemiology and risk factors for AKI.

The methodology is reasonable and clear-cut, the results and conclusions are predominantly presented in a comprehensible way.

From my point of view there are only two minor suggestions for improvement:

The intention for the subgroup-analysis by the year of the study should be explained a litle bit more detailled. Why was this done? What conclusions can be drawn from the results (discussion).

In the legend of figure 4 it should be clarified, that A) shows the ORs for mortality for patients with AKI. In the original text "...A) hospital mortality and/or 90-day mortality among patients undergoing cardiac transplantation..." , this information is missing.

Author Response

Response to Reviewer#1

As acute kidney injury after heart transplantation is a severe problem with impact on mortality and outcome, the study addresses an important topic. The integration of a number of singular clinical trials into a meta-analysis can improve the understanding of epidemiology and risk factors for AKI.

The methodology is reasonable and clear-cut, the results and conclusions are predominantly presented in a comprehensible way.

From my point of view there are only two minor suggestions for improvement:

Response: We thank you for reviewing our manuscript and for your critical evaluation.

Comment #1

The intention for the subgroup-analysis by the year of the study should be explained a little bit more detailed. Why was this done? What conclusions can be drawn from the results (discussion).

Response: We appreciated the reviewer’s important comment. We agree with the reviewer and thus we have added the details on the intention for the subgroup-analysis by the year of the study as the reviewer’s suggestion. We also added the discussion on the finding in the discussion part of our manuscript. The following text has been added.

Subgroup analyses based on year of study (before and after the year of 2015) were performed to assess if there was any difference in the incidence of AKI among studies from the recent years vs. the former years.”

“In this study, we revealed that AKI and severe AKI requiring RRT among patients undergoing cardiac transplantation are fairly common (47% and 12%, respectively). In addition, the incidence rates of AKI and severe AKI requiring RRT were increasing in the recent years. AKI post cardiac transplantation is associated with increased short term (3.5-fold) and 1-year (2.3-fold) mortality. Despite advances in transplant medicine, meta-regression showed that the risks of hospital mortality (and/or 90-day mortality) or 1-year mortality among patients undergoing cardiac transplantation with AKI has not improved over time.

Comment #2

In the legend of figure 4 it should be clarified, that A) shows the ORs for mortality for patients with AKI. In the original text "...A) hospital mortality and/or 90-day mortality among patients undergoing cardiac transplantation..." , this information is missing.

Response: We agree with the reviewer and thus we corrected the legend of figure 4 as the reviewer’s suggestion. The following text has been revised.

“Figure 4. Forest plots of the included studies assessing A) hospital mortality and/or 90-day mortality among patients undergoing cardiac transplantation with AKI and B) hospital mortality and/or 90-day mortality among patients undergoing cardiac transplantation with AKI on RRT. A diamond data label serves as the overall rate from each included study (square data marker) and 95% CI.”

We greatly appreciated the reviewer’s time and comments to improve our manuscript.

Reviewer 2 Report

This well written and scientifically sound systematic review highlights the high frequency of AKI including those cases requiring renal replacement therapy during cardiac transplantation. It provides an important background for further scientific studies in this setting.

Minor remarks :

Line 140: repetition of and and

Line 181 :  I guess "fairly" instead of faily.

In the discussion, I also suggest to the authors to put in perspective their findings with long-term CKD or ESKD observed in cardiac transplant patients generally considered to be related to calcineurin inhibitors '

This well written and scientifically sound systematic review highlights the high frequency of AKI including those cases requiring renal replacement therapy during cardiac transplantation. It provides an important background for further scientific studies in this setting.

Minor remarks :

Line 140: repetition of and and

Line 181 :  I guess "fairly" instead of faily.

In the discussion, I also suggest to the authors to put in perspective their findings with long-term CKD or ESKD observed in cardiac transplant patients generally considered to be due to toxicity of calcineurin inhibitors' toxicity. Thus, the putative additive role of previous AKI has to be studied in the light of this systematic review.

Author Response

Response to Reviewer #2

This well written and scientifically sound systematic review highlights the high frequency of AKI including those cases requiring renal replacement therapy during cardiac transplantation. It provides an important background for further scientific studies in this setting.

Response: We thank you for reviewing our manuscript and for your critical evaluation.

Comment #1

Minor remarks :

Line 140: repetition of and and

Line 181 :  I guess "fairly" instead of faily.

Response: We appreciated the reviewer’s important comments. We have corrected these errors as reviewer’s suggestion.

Comment #2

In the discussion, I also suggest to the authors to put in perspective their findings with long-term CKD or ESKD observed in cardiac transplant patients generally considered to be due to toxicity of calcineurin inhibitors' toxicity. Thus, the putative additive role of previous AKI has to be studied in the light of this systematic review.

Response: We appreciated the reviewer’s important comments. We agree with the reviewer’s comment and thus added the perspective on the use of calcineurin inhibitors and CKD progession/ESKD development among cardiac transplant recipients with AKI.

Lastly, AKI is a known independent risk factor for the development of CKD (60-62) and one of important adverse effects of calcineurin inhibitor among cardiac transplant patients is the development of CKD leading to ESKD. Future studies are required if optimizing/modifying the immunosuppressive regimen among cardiac transplant patients with AKI can help prevent CKD and delay CKD progression.”

We greatly appreciated the reviewer’s time and comments to improve our manuscript.
